# A Splice Intervention Therapy for Autosomal Recessive Juvenile Parkinson’s Disease Arising from Parkin Mutations

**DOI:** 10.3390/ijms21197282

**Published:** 2020-10-01

**Authors:** Dunhui Li, May T. Aung-Htut, Kristin A. Ham, Sue Fletcher, Steve D. Wilton

**Affiliations:** 1Centre for Molecular Medicine and Innovative Therapeutics, Murdoch University, Perth 6150, Australia; Dunhui.Li@murdoch.edu.au (D.L.); M.Aung-Htut@murdoch.edu.au (M.T.A.-H.); Kristin.Ham@murdoch.edu.au (K.A.H.); s.fletcher@murdoch.edu.au (S.F.); 2Perron Institute for Neurological and Translational Science, University of Western Australia, Perth 6009, Australia

**Keywords:** antisense oligonucleotides, exon skipping, juvenile-onset Parkinson’s disease, precision medicine

## Abstract

Parkin-type autosomal recessive juvenile-onset Parkinson’s disease is caused by mutations in the *PRKN* gene and accounts for 50% of all autosomal recessive Parkinsonism cases. Parkin is a neuroprotective protein that has dual functions as an E3 ligase in the ubiquitin–proteasome system and as a transcriptional repressor of *p53*. While genomic deletions of *PRKN* exon 3 disrupt the mRNA reading frame and result in the loss of functional parkin protein, deletions of both exon 3 and 4 maintain the reading frame and are associated with a later onset, milder disease progression, indicating this particular isoform retains some function. Here, we describe in vitro evaluation of antisense oligomers that restore functional parkin expression in cells derived from a Parkinson’s patient carrying a heterozygous *PRKN* exon 3 deletion, by inducing exon 4 skipping to correct the reading frame. We show that the induced *PRKN* transcript is translated into a shorter but semi-functional parkin isoform able to be recruited to depolarised mitochondria, and also transcriptionally represses *p53* expression. These results support the potential use of antisense oligomers as a disease-modifying treatment for selected pathogenic *PRKN* mutations.

## 1. Introduction

Parkinson’s disease (PD) is one of the most common neurodegenerative diseases, affecting approximately 1% of the population over the age of 60 years [1]. The main neuropathological hallmarks of PD are the loss of dopaminergic neurons in the substantia nigra pars compacta (SNpc) and the presence of Lewy body and Lewy neurites (α-synuclein polymers) [2] in the SNpc or other regions of the brain, such as the dorsal motor nucleus of the vagus and locus coeruleus. These disturbances produce a broad spectrum of clinical manifestations, including core motor and nonmotor symptoms. Currently, the available treatments for PD merely ameliorate its symptoms. Although some compounds have demonstrated neuroprotection in preclinical studies or clinical trials, no single intervention exists to slow PD progression or modify the disease course. A “one-for-all” treatment strategy is unlikely to be viable for PD since many causative genes and molecular pathways are involved in its pathophysiology. Therefore, multiple therapeutic strategies are needed, particularly for targeted precision medicines tailored to each PD subtype or individual.

Parkin-type autosomal recessive juvenile PD (ARJP) is a form of PD caused by *PRKN* mutations and accounts for ~50% of autosomal recessive Parkinsonism and ~15% of early-onset sporadic PD [3]. The ARJP subtype is clinically characterised by early symmetrical onset, dystonia, and hyperreflexia. Patients with Parkin-type ARJP generally show a positive response to levodopa; however, motor fluctuations and levodopa-induced dyskinesia symptoms are frequent during the disease course. Since Lewy bodies and Lewy neurites are rarely seen in Parkin-type ARJP, this PD subtype is believed to be a distinct clinical entity [4] and shows different pathogenesis compared to the α-synuclein pathology [5,6].

The *PRKN* protein, parkin, functions as a RING-Between-RING E3 ligase in the ubiquitin–proteasome system and participates in the mitochondrial quality control network to maintain mitochondrial homeostasis [7]. Parkin deficiency leads to mitochondrial swelling, cytochrome *c* release, and caspase activation [8], and may indicate the central role of mitochondrial dysfunction in the pathogenesis of Parkin-type ARJP. On the other hand, parkin function deficiency compromises its transcriptional repression of *p53* and consequently mediates programmed neuronal cell death and neurodegeneration [9,10]. Genotype–phenotype studies on Parkin-type ARJP patients show that patients with genomic deletion of both exons 3 and 4 in *PRKN* present with milder symptoms than patients with the out-of-frame deletion of exons 3 or 4 alone [11,12]. These observations suggest that the parkin isoform missing domains encoded by exons 3 and 4 is semi-functional, and thus provides justification that inducing a *PRKN* mRNA isoform missing exons 3 and 4 may offer a therapeutic avenue for those ARJP patients with disease-causing mutations involving either exon 3 or 4.

Antisense oligonucleotides (AOs) are synthetic nucleic acid analogues that can be designed to anneal to specific RNA sequences. Depending upon the AO chemistry and target site, the AO may promote RNA degradation, alter mRNA translation, or alter pre-mRNA splicing [13]. Several splice-switching AOs are now in the clinic, with the accelerated FDA approvals of exon-skipping drugs Exondys51^®^ and Vyondys53^®^ for the treatment of Duchenne muscular dystrophy (DMD) and exon-including drug Spinraza^®^ for the treatment of spinal muscular atrophy [14]. In this study, we examined the potential of AO-induced exon skipping as a disease-modifying treatment for Parkin-type ARJP arising from amenable mutations in *PRKN*. We designed and evaluated 2′-O-methyl phosphorothioate (2′OMe PS) AOs to elicit efficient exon 4 skipping in Parkin-type ARJP patient-derived dermal fibroblasts carrying a genomic deletion of exon 3. The lead AO sequence from this trial was sourced as a phosphorodiamidate morpholino oligomer (PMO) that subsequently demonstrated improved skipping of exon 4 and restoration of some parkin function in patient-derived cells. Our data, therefore, support the therapeutic potential of AO-induced exon skipping in Parkin-type ARJP patient-derived cells carrying amenable *PRKN* mutations.

## 2. Results

### 2.1. Design of Antisense Oligomers to Induce PRKN Exon 4 Skipping

*PRKN* is one of the largest genes, consisting of 12 exons spanning 1.8 Mb to generate the *PRKN* mRNA transcript of 4167 bases. The *PRKN* exon map and protein structure are shown in Figure 1A and 1B. Genomic deletion of *PRKN* exon 3 results in an out-of-frame transcript and a prematurely truncated nonfunctional protein (Figure 1C). In contrast, the deletion of *PRKN* exons 3 and 4 leads to a shorter in-frame transcript and an internally truncated parkin protein that preserves most of the functional domains (Figure 1D).

Splice modulating antisense compounds mediate exon skipping through steric blocking of motifs involved in exon recognition and pre-mRNA processing, such as the acceptor/donor splice sites or exon splicing enhancers. To induce skipping of *PRKN* exon 4, we designed AOs to target the acceptor or donor splice sites, as well as intraexonic splicing enhancers, as predicted by SpliceAid [15] (http://www.introni.it/splicing.html; Figure 1E). Four AOs (Table 1) were designed and synthesised as 2′OMe PS AOs for an initial evaluation in patient and healthy control-derived fibroblasts.

### 2.2. Antisense Oligonucleotide-Induced Exon 4 Skipping to Restore the PRKN Reading Frame in Patient-Derived Fibroblasts

The 2′OMe PS AOs targeting *PRKN* exon 4 were transfected as cationic lipoplexes into the typical ARJP patient-derived fibroblasts. Patient compound heterozygous *PRKN* mutations: exon 3 deletion on one allele and c.719 C > T missense mutation on the other were confirmed by Sanger sequencing (Appendix A). The *PRKN* transcripts were assessed by RT-PCR across exons 1–6 to assess levels of exon 4 skipping. In the untreated samples (Figure 2A), the amplicons representing the transcript with the exon 3 genomic deletion are in slight excess of the full-length product, presumably due to preferential amplification of the shorter transcript. Nonsense-mediated decay does not appear to impact on transcript abundance in the patient cells. The control AO that does not anneal to any transcript did not affect *PRKN* exon 4 splicing in patient-derived fibroblasts. All AOs targeting the *PRKN* transcript induced variable levels of exon 4 skipping, between 5% and 20% (Figure 2A). The shorter amplicons were identified by Sanger sequencing (Figure 2B) and found to be missing exon 4 from the full-length allele carrying the c.719 C > T missense mutation, or exons 3 and 4 from the allele carrying the exon 3 genomic deletion. In the initial screening, AO2 PRKN_H4A (+07 + 33) showed the highest exon skipping efficiency of approximately 30% at 25 nM (Figure 2A). Further refinement of AO2 was undertaken by “microwalking” additional AOs (AOs 5–7) with overlapping sequences of varying lengths (Figure 1E and Table 1). The subsequent evaluation identified AO7 PRKN_H4A (+13 + 34) as the most efficient, with approximately 50% skipping of exon 4 induced at 25 nM (Figure 2C,D).

### 2.3. Morpholino Oligomer-Induced PRKN Exon 4 Skipping and Production of a Shorter Functional Parkin Isoform

The oligomer that induced the most efficient exon 4 skipping, AO7, was subsequently sourced as a PMO from Gene Tools, LLC (Philomath, OR, USA). A control PMO from Gene Tools (GTC), which has no specific target and very little biological activity in the cells tested, was included as a negative transfection control. The PMOs were delivered into *PRKN* patient-derived fibroblasts using EndoPorter^®^ (Gene Tools, LLC) and assessed for exon skipping efficiency using RT-PCR analysis (Figure 3A). A low level of cell death was observed (data not shown) in all PMO-treated samples. Two amplicons of 711 bp and 470 bp, corresponding to the skipping of exon 4 from the full-length transcript and from the exon 3-deleted transcript, respectively, were observed after amplification of the *PRKN* transcript, indicating robust exon 4 skipping from both *PRKN* transcripts induced by the PMO H4A (+13 + 34). Low levels of endogenous *PRKN* exon 4 skipping were also observed in untreated or GTC-treated samples. Because skipping of *PRKN* exon 4 from Δ3 transcripts is expected to restore the *PRKN* reading frame, we analysed the parkin protein expression in the PMO-treated and untreated samples and found a shortened parkin protein that was most likely translated from ∆3 + 4 transcripts (Figure 3B). No truncated protein was observed in the GTC-treated or untreated patient-derived fibroblasts, even with a longer exposure (Appendix A).

We then evaluated the function of this truncated parkin isoform. It has been reported that functional parkin protein is recruited to the outer membrane of depolarised mitochondria to initiate mitophagy [16,17]. Patient fibroblasts were treated with 20 μM PMO H4A (+13 + 34) for 72 h, followed by depolarisation of the mitochondria by 50 μM carbonyl cyanide m-chlorophenyl hydrazone (CCCP) treatment for two hours. After 50 μM CCCP treatment, cytoplasmic healthy-type parkin protein colocalises with depolarised mitochondria in the perinuclear region, as shown by the colocalisation of parkin and the mitochondria outer membrane protein, Tomm20 (Figure 3C). A similar pattern of parkin protein location was shown in PD patient-derived fibroblasts treated with PMO but was not evident in the GTC PMO-treated or untreated PD patient fibroblasts. Additional figures indicating colocalisation of the truncated functional parkin protein and depolarised mitochondria are shown in Appendix A.

In addition to maintaining mitochondrial homeostasis, parkin protein is also known to regulate *p53* transcription. Therefore, we have also investigated the changes in *p53* expression after *PRKN* exon 4 skipping in the patient-derived fibroblasts. The parkin patient RNA samples were extracted and cDNA was synthesised for the real-time qPCR analysis to compare the *p53* expression levels with or without the PMO treatment. As shown in Figure 3D, there was an approximate 40% reduction of the *p53* expression after treatment with 20 μM PMO H4A (+13 + 34) for 72 h, compared to the GTC PMO-treated and untreated patient-derived fibroblasts. Dose-dependent repression of *p53* expression also corresponded to the exon skipping levels observed by RT-PCR.

### 2.4. 2′ OMe PS AO Cocktail-Mediated PRKN Exon 3 Skipping in Healthy Human Fibroblasts

After successful induction of exon 4 skipping in the patient-derived fibroblasts with genomic deletion of exon 3, we postulated that excising exon 3 or both exon 3 and 4 at the same time could be a potential therapy for ARJP patients carrying a genomic deletion of exon 4 or those with intraexonic protein-truncating or missense mutations in exons 3 or 4, respectively. Thus, we designed and synthesised AOs to skip *PRKN* exon 3 to be used in conjunction with PMO H4A (+13 + 34). Fourteen 2′OMe PS AO sequences annealing to exon 3 and the exon–intron boundaries (Figure 4) were transfected into healthy human fibroblasts. The sequences of AOs are shown in Table 2. None of the AOs individually induced exon 3 skipping when analysed by RT-PCR and electrophoresis (data not shown), while some combinations of two AOs transfected into the healthy human fibroblasts as cocktails did induce minimal exon 3 skipping. Combinations of three nonoverlapping AOs were then transfected into the healthy human fibroblasts and we observed that cocktails 3 and 4 induced approximately 20% skipping of exon 3. A slightly higher level of exon 3 skipping (35%) was induced when four AOs were combined as cocktail 5 (Figure 5).

## 3. Discussion

It is estimated that the human genome contains over 230,000 exons [18]; however, it is becoming apparent that the definition of the terms “gene”, “exon” and “intron” are becoming more flexible. The “average” human gene consists of eight to nine exons spread across about 30 kb and generates a mature mRNA of approximately 1 kb. Hence, pre-mRNA processing is a fundamental step during expression for more than 95% of human genes. Increased genetic plasticity arises from alternative splicing, whereby different permutations of exons and sometimes intronic sequences may be combined in a tissue-specific and/or developmental manner [19]. It has also become apparent that not all exons need to be retained in a gene transcript for the near-normal function of the encoded gene products.

Duchenne muscular dystrophy, the most common and severe form of childhood muscle wasting, arises from mutations that lead to premature termination of translation [20]. Becker muscular dystrophy, a variable but milder form of muscular dystrophy, also arises from dystrophin gene lesions but these are generally in-frame deletions in the central rod domain. The characterisation of mild, or asymptomatic cases of Becker muscular dystrophy arising from the loss of an in-frame dystrophin exon(s) indicates that many of these exons are dispensable [21]. While it should not be surprising that there are redundant exons in a large multiexon gene such as dystrophin, examples of a milder phenotype despite the loss of one or more exons in other genes, including *PRKN*, have also been reported.

Clinical genotype–phenotype studies have shown that PD patients carrying a deletion of both exons 3 and 4 from *PRKN* have milder symptoms compared to patients with a parkin deficiency [11,12]. Such an explanation for the basis of disease variability in PD opens up new therapeutic strategies for Parkin-type ARJP by changing a severe disease phenotype into a milder form through redirecting *PRKN* pre-mRNA processing to induce an isoform missing exons 3 and 4. In this proof-of-concept study, we demonstrated that exon skipping AOs could efficiently remove *PRKN* exon 4 and restore the reading frame in a *PRKN* transcript disrupted by the genomic deletion of exon 3. The induced in-frame *PRKN* transcript missing exons 3 and 4 is translated into an internally truncated isoform that appears to be functional. We have shown that excising both *PRKN* exons 3 and 4 has little effect on parkin protein function in vitro, providing yet another example of potentially “redundant” exons as demonstrated in other genes, such as *DMD*.

The importance of parkin in the ubiquitin–proteasome system as an E3 ligase and its implications in PD pathogenesis has recently driven substantial research into its structure [22]. The six domains of parkin form an autoinhibitory conformation where catalytic RING2 is blocked by RING0 [23]. Upon phosphorylation by PTEN-induced kinase 1, parkin undergoes a conformational change to expose the RING2 domain and undertake its E3 ligase activity [24]. Several parkin constructs without the Ubl domain, the first 95 aa, or the Ubl and the following linker domain, were not compromised in their catalytic activity, compared to normal parkin [25]. The internally truncated parkin protein, missing 25% of the Ubl domain, the entire linker domain, and 44% of the RING0 domain encoded by exon 3 and 4, was seen to colocalise with Tomm20 on depolarised mitochondria to initiate mitophagy in patient cells, which is consistent with previous studies [16,26]. Another essential function of parkin is to repress *p53* transcriptionally. Irrelevant to its E3 ligase activity, parkin RING1 domain, or the RING1 plus the following IBR domains are shown to be essential for *p53* repression [10]. The PMO-induced parkin isoform that has an intact RING1 and the remaining domains were demonstrated to reduce the level of *p53* mRNA by 40%. As a 2-fold difference in *p53* mRNA expression level was detected between the healthy population and Parkin-type ARJP [10], reducing 40% of *p53* expression in patients is expected to bring the *p53* level close to that found in the healthy cohort. These results confirm our hypothesis on the dual role of neuroprotective parkin and support the current knowledge of parkin protein structure and function as well as the clinical genotype–phenotype correlation.

A large number of genetic variations, including missense, splice site mutations or deletions have been reported across the entire *PRKN* gene [27]. A contributing factor to the frequency of *PRKN* deletion mutations is *PRKN’s* location within a chromosomal fragile site, FRA6E (6q26) [27,28]. Genomic deletions in *PRKN* are responsible for 50% of familial early-onset PD cases, with exon 3 and/or 4 deletions (fragile hotspot in 6q26) making up half of these cases [29]. Therefore, in addition to developing an exon 4 skipping compound to address *PRKN* exon 3 deletions, we designed and evaluated AOs to excise exon 3 in healthy human fibroblasts. However, unlike *PRKN* exon 4 that was relatively straight-forward to excise during pre-mRNA processing in patient-derived fibroblasts, none of the single AOs targeting exon 3 that we tested induced any exon skipping. We had previously reported that combinations of AOs were sometimes more effective [30,31]; however, only a few preparations out of 44 different two-AO cocktails induced a maximum of about 10% exon 3 skipping. We have observed considerable variation in exon skipping efficiencies across several genes [30,32], and some exons were found to be particularly challenging to excise from the mature mRNA. It is speculated that, because massive introns flank *PRKN* exon 3, one would expect the presence of strong splice enhancers to promote its inclusion. As the evaluation of exon 3 skipping AOs in this study was performed initially in healthy cells, it is possible that those individuals with genomic deletions of *PRKN* exon 4 may be more amenable to exon 3 skipping.

The development of antisense oligomer therapeutics for Parkin-type ARJP is still at an early stage, and PD patient dermal fibroblasts have thus far provided a useful model for characterising *PRKN* mutations and their resulting cumulative cellular damage in patients [33]. However, further examination of the therapeutic potential of AOs must be undertaken in patient dopaminergic neurons that could be differentiated from patient dermal fibroblast-derived induced pluripotent stem cells [34]. With further studies in PD subtyping and characterisation according to specific genetic-molecular pathogenesis, we may gain additional insights into disease-modifying therapies for Parkin-type ARJP and develop personalised therapies for amenable mutations.

## 4. Materials and Methods

### 4.1. Antisense Oligomer Design and Synthesis

Antisense oligomers were designed to target *PRKN* exon 4 acceptor or donor splice sites, and exon splicing enhancers, as predicted by the online splice prediction tool: SpliceAid [15] (http://www.introni.it/splicing.html). Antisense oligomers composed of 2′-O-methyl modified bases on a phosphorothioate backbone were synthesised in-house on an Expedite 8909 Nucleic Acid synthesiser with reagents from Azco Biotech (Oceanside, CA, USA). The nomenclature of AOs is according to that reported by Mann et al. [35,36], and provides a description of annealing coordinates within the targeted exon. After initial screening, the most promising AO sequences were optimised by shifting the annealing sequence five bases upstream or downstream. The optimal AO sequence was sourced as a PMO by Gene Tools, LLC (Philomath, OR, USA).

### 4.2. Cell Propagation

Parkin-type ARJP patient dermal fibroblasts and healthy human primary dermal fibroblasts were derived from skin biopsies (Human Ethics Committee of Murdoch University approval 2013/156). Cells were proliferated in Dulbecco’s Modified Eagle Medium (DMEM, Gibco, Life Technologies, Melbourne, Australia) supplemented with 15% fetal bovine serum (FBS, Serana, Bunbury, Australia) in 75 cm^2^ flasks at 37 °C in a 5% CO_2_ atmosphere before seeding into 24-well plates at the cell density of 15,000 per well for RNA extraction, or 10,000 per cover slip for immunofluorescence studies. Cells were cultured in 10% FBS DMEM for 24 h before transfection.

### 4.3. Transfection

Cells were transfected with 2′OMe PS AO: Lipofectamine 3000 (L3K, 3 µL, Life Technologies) lipoplexes in Opti-MEM (Life Technologies) according to the manufacturer’s instructions, at concentrations of 100, 50, and 25 nM for initial screening. PMO solutions were warmed for 5 min at 37 °C before being transfected into cells with EndoPorter^®^ (Gene Tools) according to the manufacturer’s instructions at the concentration of 20 μM. Culture medium was replaced with fresh 10% FBS DMEM before adding the desired volume of PMO and 3 µL of EndoPorter^®^ for every 500 µL culture medium. Cells were incubated for 24 h for 2′OMe AO transfections and 72 h for PMO transfections before harvesting for RNA extraction.

### 4.4. RNA Extraction and PCR

The MagMax^TM^ 96 total RNA isolation kit (Life Technologies) was used to extract total RNA from cultured cells, according to the manufacturer’s guidelines. RT-PCR was performed using approximately 50 ng of total RNA with Superscript III One-Step RT-PCR System with Platinum^®^ Taq DNA Polymerase (Thermo Fisher Scientific, Waltham, MA, USA). The PCR conditions used were 55 °C for 30 min, 94 °C for 2 min, followed by 32 cycles of 94 °C for 30 s, 60 °C for 1 min, and 68 °C for 2 min. Primers (PRKN_Ex1F: 5′-TGGAGGATTTAACCCAGGAG-3′; PRKN_Ex3F: 5′-ATGAATGCAACTGGAGGCGA-3′; PRKN_6R: 5′-GACGTCTGTGCACGTAATGC-3′) were designed using NCBI primer blast [37] and supplied by Integrated DNA Technologies, Singapore. RT-PCR products were resolved on a 2% agarose gel in TAE buffer using a 100 bp DNA ladder (Thermo Fisher Scientific) as the size standard. The sizes of full-length exon 4, exon 3, and exons 3 + 4 excised *PRKN* mRNA transcript products are 833 bp, 711 bp, 592 bp, and 470 bp, respectively. RT-PCR products of interest were band-stabbed [38], or excised for purification by either Diffinity Rapid Tip^TM^ (Diffinity Genomics, West Chester, PA, USA) or NucleoSpin^®^ Gel and PCR Clean-up (Scientifix Life, Melbourne, Australia), respectively, according to the manufacturer’s instructions. Purified PCR products were sent to the Australian Genome Research Facility Ltd. (Nedlands, Australia) for Sanger sequencing.

### 4.5. cDNA Synthesis and Real-Time PCR

The SuperScript^TM^ IV First-Strand Synthesis System (Thermo Fisher Scientific) was used to synthesise cDNA. A 5 μL volume of the total RNA was mixed with 0.5 μL random hexamers (100 ng/μL) and 1 μL dNTPs (5 mM) and heated to 65 °C for 5 min, followed by 5 min on ice. A 2 μL volume of 5× first-strand buffer, 0.5 μL 0.1 M DTT, 0.5 μL RNase Out, and 0.5 μL Superscript IV reverse transcriptase was added into the mixture before the cDNA synthesis using the following condition: 23 °C for 10 min, 50 °C for 10 min and 80 °C for 10 min. Real-time PCR (qPCR) reactions were performed by C1000^TM^ Thermal Cycler (Bio-Rad) using Fast SYBR^TM^ Green Master Mix (Thermo Fisher Scientific). Each PCR reaction (10 μL) contained 5 μL 2× SYBR Green Master Mix, 400 nM *p53* primers (p53_Ex1/2qF: 5′-GCTTCCCTGGATTGGCAGC-3′; p53_Ex2qR: 5′-GACGCTAGGATCTGACTGCG-3′) or 500 nM TBP primers (TBP_Ex1/2qF: 5′-TCTTTGCAGTGACCCAGCATCAC-3′; TBP_Ex2qF: 5′-CCTAGAGCATCTCCAGCACACTCT-3′), 3 μL cDNA, and H_2_O (if needed to make up to 10 μL volume). All qPCR reactions were conducted in triplicate. The cycling conditions consisted of one single step of 95 °C for 1 min followed by 40 cycles (95 °C for 20 s, 60 °C for 20 s and 72 °C for 20 s). A final melting program ranging from 65 to 95 °C with a heating rate of 0.5 °C per 10 s was performed to create a melt curve. Negative controls with no cDNA samples were prepared. Standard curves for both *p53* and *TBP* were made to test the efficiency of the primers as well as for the qPCR analysis. The Ct value was analysed by using Bio-Rad CFX manager 2.1 and the *PRKN* expression level relative to *TBP* was calculated using the ΔΔCt method.

### 4.6. Western Blot

Western blotting was performed using a protocol derived from Cooper et al. and Nicholson et al. [39,40,41]. Cells were harvested and resuspended in treatment buffer (100 μL/4.5 mg wet pellet weight) consisting of 125 mmol/L Tris-HCl pH 6.8, 15% sodium dodecyl sulphate, 10% glycerol, 0.5 mmol/L phenylmethylsulfonyl fluoride, 50 mmol/L dithiothreitol, bromophenol blue (0.004% w/v), and a protease inhibitor cocktail (3 μL/100 μL of treatment buffer, Sigma-Aldrich, Castle Hill, Australia). Samples were vortexed briefly, sonicated for 1 s, 4–8 times at a setting of 30/100 on an ultrasonic processor (Sonics, Newtown, CT, USA) and heated at 95 °C for 5 min. Total protein concentration was determined by Pierce BCA protein assay kit (Thermo Fisher Scientific) without bromophenol blue and dithiothreitol. Approximately 15 µg of total protein was loaded for each sample on the NuPage 4–12% Bis/Tris gradient gel (Life Technologies) with 7 µL of Kaleidoscope^TM^ (Bio-Rad) and 3 µL of MagicMark XP (Thermo Fisher Scientific). Electrophoresis was carried out at 200 V for 1 h in 1 × NuPage MOPS SDS running buffer. The fractionated protein was transferred to a methanol pre-treated polyvinylidene difluoride (PVDF) membrane at 350 mA for 1 h in the Towbin buffer. The membrane was blocked with 5% skim milk in TBS-T (0.1% Tween20 in 1× TBS) for 1 h before incubation with MAB5512 (1:500, Merck, Castle Hill, Australia) and anti-β-tubulin antibody (1:1000, Developmental Studies Hybridoma Bank, USA) at room temperature for 1 h. Horseradish peroxidase goat anti-mouse (1:10,000) and goat anti-rabbit (1:10,000) secondary antibodies (Dako Agilent, Santa Clara, CA, USA) were incubated for 1 h at room temperature before images were detected using the Fusion FX system (Vilber Lourmat, Marne-la-Vallée, France) and Fusion-Capt software.

### 4.7. Immunofluorescence

Three days after the 20 μM PMO transfection, 10,000 patient fibroblasts seeded onto glass cover slips were treated with 50 μM carbonyl cyanide m-chlorophenyl hydrazone (CCCP) for 2 h to depolarise the mitochondria. The glass cover slips were collected and fixed in acetone/methanol (1:1) on ice and 20% goat serum in PBS-T (0.2% Triton-X in 1× PBS) was used to block the slides for 1 h. Parkin was probed with MAB5512 (1: 200, Merck) for 1 h and Alexa Fluor mouse 488 (1:400, Thermo Fisher Scientific) for a further hour. Tomm20 was probed with HPA011562 (1:100, Sigma-Aldrich) at room temperature for 1 h and Alexa Fluor rabbit 568 (1:400, Thermo Fisher Scientific) for another hour. Nuclei were stained with Hoechst (1:160, Sigma-Aldrich). Images were captured and analysed using the Nikon Eclipse 80i microscope with the NIS elements program. 

## 5. Patents

An Australian Provisional Patent Application 2020900220 was lodged on the 28th January 2020.

## Figures and Tables

**Figure 1 ijms-21-07282-f001:**
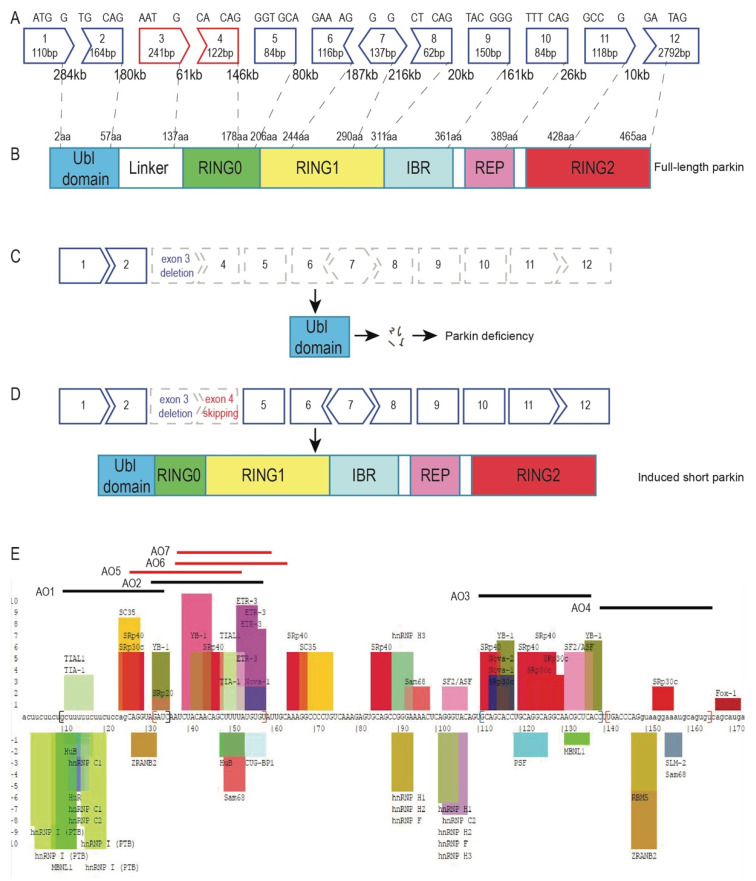
*PRKN* exon map, parkin protein structure, and the strategy to restore parkin function. (**A**) *PRKN* is a large gene spanning around 1.8 Mb of DNA with 12 exons separated by huge introns. (**B**) Parkin is a RING-between-RING E3 ligase that mainly consists of the ubiquitin-like domain, RING0, RING1, in-between-ring motif (IBR), repressor element of parkin (REP), and the RING2 domain. (**C**) A *PRKN* exon 3 genomic deletion disrupts the reading frame and causes parkin deficiency. (**D**) Skipping exon 4 in patient-derived fibroblasts carrying an exon 3 deletion restores the *PRKN* reading frame. The induced transcript can be translated into a shorter protein missing a small part of the ubiquitin-like domain, the linker, and a portion of the RING0 domain, but preserving the catalytic RING domains. (**E**) Exon splicing enhancers and silencers were predicted by SpliceAid [15]. The original image of splicing motifs and sequence information can be found on the SpliceAid website (http://www.introni.it/splicing.html). Initial splice-switching *PRKN* exon 4 AO sequences (AO 1–4) were designed to target the exon splicing enhancers or exon–intron boundaries. Design of “microwalking” AOs (AO 5–7) was refined from the AO 2 sequence. The splicing enhancer and silencer motifs are shown as coloured bars above and below the sequence, respectively. The height of the bars represents the predicted strength of the motifs. The intronic sequences are shown in lower case and the exonic sequences as upper case.

**Figure 2 ijms-21-07282-f002:**
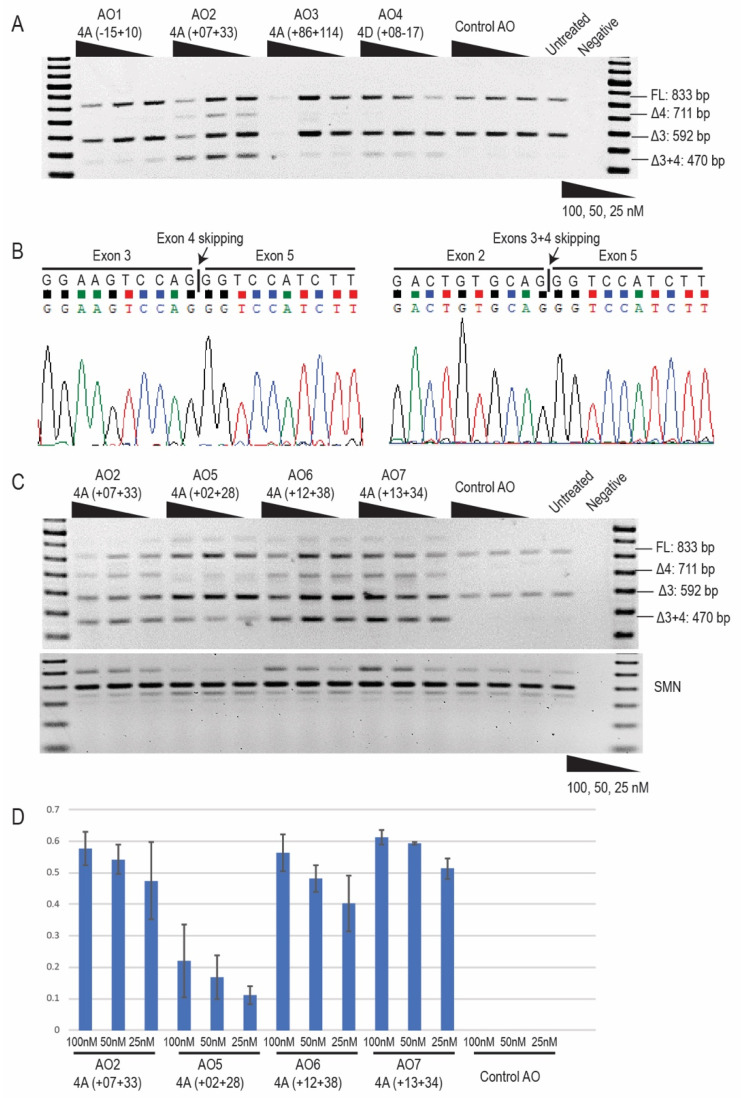
*PRKN* transcript analysis following 2′-O-methyl PS AO transfection. (**A**) RT-PCR analysis of *PRKN* transcripts from Parkinson’s patient-derived fibroblasts with genomic deletion of exon 3, following transfection with *PRKN* exon 4 targeting 2′OMe PS AOs at concentrations of 100, 50, and 25 nM. The control AO that does not anneal to any transcript was used as a negative control. A 100 bp DNA ladder was used to indicate amplicon sizes, and an RT-PCR no-template negative control was loaded in the last lane. (**B**) Sanger sequencing identified skipping of exon 4 on the allele carrying the missense mutation and the absence of exons 3 + 4 on the allele with *PRKN* exon 3 genomic deletion. (**C**) RT-PCR analysis of *PRKN* transcripts after the transfection of “microwalking” AOs. The original AO sequence, AO2, was included for comparison of exon skipping efficiencies. *SMN* transcript was used as a house-keeping gene and normalization of *PRKN* transcripts. (**D**) Densitometric analysis of the Δ3 + 4 band showed a dose-dependent response to all of the AOs and indicated the AO sequence that induced the highest level of exon skipping. FL; full-length.

**Figure 3 ijms-21-07282-f003:**
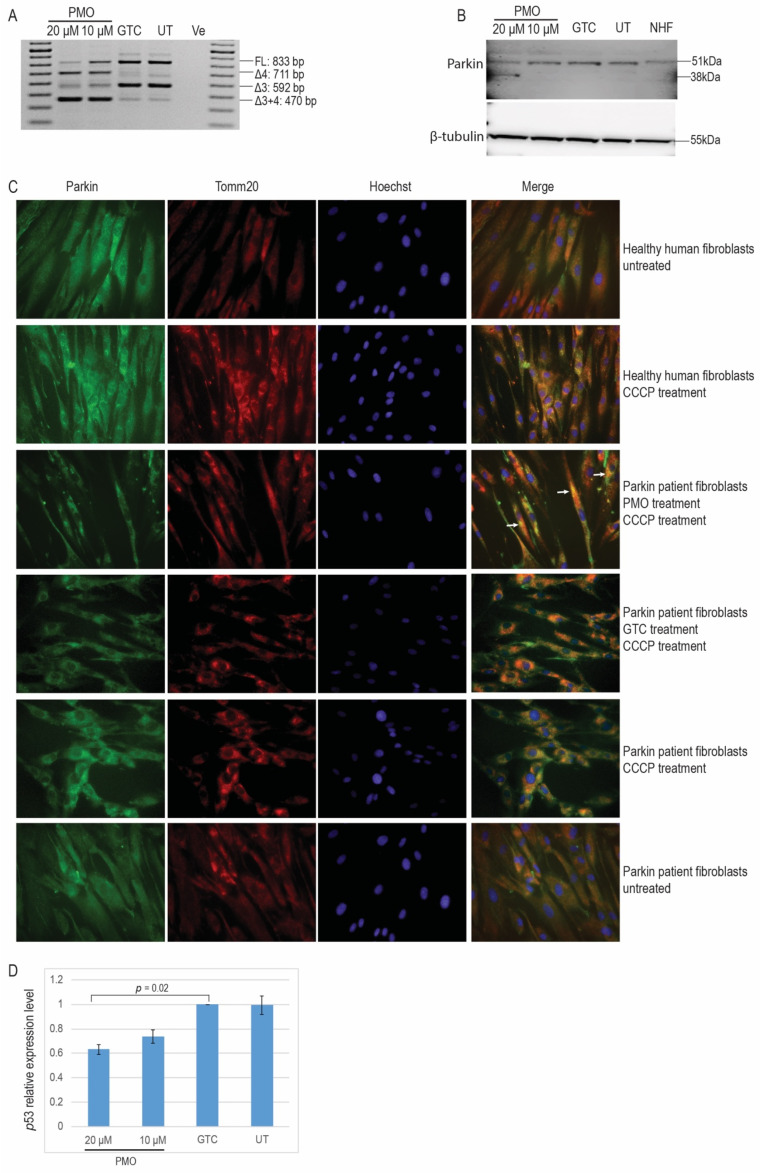
*PRKN* transcript analysis following PMO treatment and functional assessment of the induced shorter parkin isoform. (**A**) RT-PCR analysis confirmed the skipping of *PRKN* exon 4 after PMO treatment in patient-derived fibroblasts at a concentration of 20 μM. The Gene Tools control sequence was used as a negative control. (**B**) Western blot analysis of parkin protein showed the induction of a shorter parkin protein after PMO transfection at 20 μM in patient-derived fibroblasts. (**C**) Immunofluorescence labelling of the parkin protein and Tomm20, a mitochondria outer membrane protein. Wild-type parkin protein colocalises with the depolarised mitochondria of healthy human fibroblasts treated with CCCP. A similar pattern was observed in the PMO-treated patient fibroblasts, while the defective parkin did not colocalise with the impaired mitochondria in the Gene Tools control PMO-treated and untreated patient fibroblasts. Arrows indicate the proper location of truncated functional parkin protein to the depolarised mitochondria. (**D**) Real-time PCR analysis of *p53* transcript. The *p53* expression level was quantitated using the ΔΔCt calculation. Average normalised *p53* mRNA expression levels for each sample were calculated and are presented. Compared to the GTC PMO-treated sample, a decrease of approximately 40% in the level of *p53* mRNA was observed after treatment of the cells with 20 μM PMO designed to skip exon 4 (*p* = 0.02), *n* = 3. FL; full-length; GTC; Gene Tools control; UT; untreated; NHF: normal human fibroblasts; CCCP: carbonyl cyanide m-chlorophenyl hydrazone.

**Figure 4 ijms-21-07282-f004:**
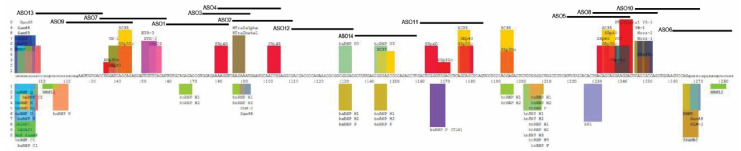
Exon splicing enhancers, silencers, and AO binding sites for *PRKN* exon 3. Exon splicing enhancers and silencers were predicted by SpliceAid [15]. Original images of splicing motifs and sequence information can be found on the SpliceAid website (http://www.introni.it/splicing.html). Fourteen AO sequences were designed to anneal to the exon 3 splicing enhancers and the exon–intron boundaries.

**Figure 5 ijms-21-07282-f005:**
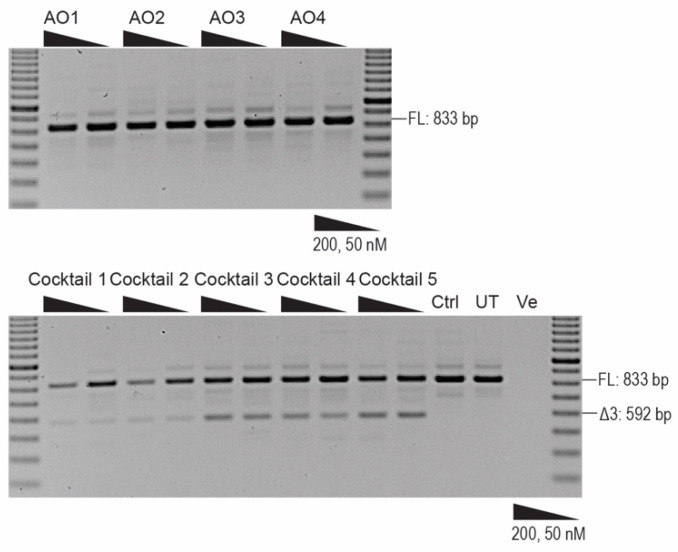
RT-PCR analysis of *PRKN* transcript following transfection of AO cocktails to skip exon 3 in healthy human fibroblasts. The *PRKN* transcript was amplified using RT-PCR to analyse exon 3 skipping. A 100 bp DNA ladder was used to indicate the sizes of the PCR products. Cocktail 1 is made up of H3A (+09 + 35) and H3A (+36 + 60); cocktail 2 is made up of H3A (+09 + 35) and H3A (+62 + 86); cocktail 3: H3A (+09 + 35), H3A (+36 + 60) and H3A (+62 + 86); cocktail 4: H3A (+09 + 35), H3A (+36 + 60) and H3A (+55 + 81); cocktail 5: H3A (+09 + 35), H3A (+36 + 60), H3A (+55 + 81), and H3A (+62 + 86). FL: full-length transcript; Ctrl: control AO; UT: untreated. Ve; RT-PCR with no-template.

**Table 1 ijms-21-07282-t001:** Antisense oligonucleotides designed and used to induce *PRKN* exon 4 skipping.

AO Number	AO Nomenclature	Sequence 5′→3′	Length (bp)
1	PRKN_H4A (−15 + 10)	GAUCUACCUGCUGGAGAAGAAAAAG	25
2	PRKN_H4A (+07 + 33)	ACACAUAAAAGCUGUUGUAGAUUGAUC	27
3	PRKN_H4A (+86 + 114)	GGUGAGCGUUGCCUGCCUGCAGGUGCUGC	29
4	PRKN_H4D (+08 − 17)	ACACUGCAUUUCCUUACCUGGGUCA	25
5	PRKN_H4A (+02 + 28)	UAAAAGCUGUUGUAGAUUGAUCUACCU	27
6	PRKN_H4A (+12 + 38)	GCAAUACACAUAAAAGCUGUUGUAGAU	27
7	PRKN_H4A (+13 + 34)	UACACAUAAAAGCUGUUGUAGA	22

**Table 2 ijms-21-07282-t002:** Antisense oligonucleotides designed and used to induce *PRKN* exon 3 skipping.

AO Number	AO Nomenclature	Sequence 5′→3′	Length (bp)
1	PRKN_H3A (+09 + 35)	AUGUGAACAAUGCUCUGCUGAUCCAGG	27
2	PRKN_H3A (+205 + 230)	CUGGUGGUGAGUCCUUCCUGCUGUCA	26
3	PRKN_H3A (−05 + 22)	UCUGCUGAUCCAGGUCACAAUUCUGUU	27
4	PRKN_H3A (−05 + 21)	AUUACCUGGACUUCCAGCUGGUGGUG	26
5	PRKN_H3A (+137 + 161)	CUGAGCUGCUGAGGUCCACCCGAGU	25
6	PRKN_H3A (+36 + 60)	ACCUUUUCUCCACGGUCUCUGCACA	25
7	PRKN_H3A (+62 + 86)	UCGCCUCCAGUUGCAUUCAUUUCUU	25
8	PRKN_H3A (+50 + 69)	CAUUUCUUGACCUUUUCUCC	20
9	PRKN_H3A (+55 + 81)	CCAGUUGCAUUCAUUUCUUGACCUUU	26
10	PRKN_H3A (+195 + 219)	GUCCUUCCUGCUGUCAGUGUGCAGA	25
11	PRKN_H3D (+05 − 20)	UCUUAGAGCAUUCCAAUUACCUGGA	25
12	PRKN_H3A (−17 + 09)	GUCACAAUUCUGUUUGGGAGCAAGGU	26
13	PRKN_H3A (+86 + 110)	GACGACCCCAGAAACGCGGCGGGAG	25
14	PRKN_H3A (+111 + 135)	GCUGUGAGCGGGAGCCCCAGAGCUU	25
15	PRKN_H3A (+162 + 186)	UCCUCCCAGGAGACUCUGUGGGGCU	25

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
