# Peer review of "A Splice Intervention Therapy for Autosomal Recessive Juvenile Parkinson’s Disease Arising from Parkin Mutations"

_ijms, 2020, doi:10.3390/ijms21197282_

Round 1

Reviewer 1 Report

The research article ijms-899772 by Li et al proposes specific antisense oligomers (AO) as a potential treatment for a sub-type of Parkinson’s disease caused by PRKN exon-3 deletion. It has been shown that removal of exon-3 alters the wild type translational reading frame and is associated with severe clinical symptoms whereas deletion of both exon-3 and 4 preserves the reading frame and causes less severe clinical symptoms. Li et al hypothesize that the severe phenotype due to exon-3 deletions can potentially be lessened by introducing AO-mediated exon-4 skipping in patients carrying the naturally according exon-3 deletion, a clever idea that comes with the inherent limitations and disadvantages of AO-mediated treatment approaches. To evaluate this idea, Li et al performed in vitro experiments on fibroblast cells from a patient affected with autosomal recessive juvenile onset Parkinson due to PRKN exon-3 deletion on one allele and a missense mutation on the other allele. Authors show that, in vitro, they were able to “revert” the wild type reading frame in exon-3 deletion-containing patient fibroblasts by using AO-mediated exon-4 skipping. They also show that the “corrected” transcript produces the expected internally truncated parkin protein capable of performing molecular functions to some extent. Overall, it is a well-designed and thought-out study. The experiments include appropriate positive and negative controls, and results are presented as a well written report. In this reviewer’s opinion this article should be accepted for publication. Here are a few minor suggestions for author’s consideration.

Line 116: This is the first-time manuscript discusses about the missense mutation, at least in the draft provided to this reviewer. Up until this point, only exon-3 deletion is referred. I understand later in the methods section authors have explained that the patient is compound heterozygous for exon-3 deletion and a missense mutation, but it would be helpful for readers to provide this information in the results section before line 116.

Figure 2A: The legend for concentration gradient shown at the bottom right corner of the gel should be moved to someplace else so that it is clearly visible whether or not the Control AO, Untreated, and Negative samples have any band corresponding to exon 3+4 skipping (it is possible, at least theoretically, that the genomic deletion containing exon 3 and surrounding intronic sequences includes distant splice enhancers for exon-4, thus causing some exon-4 skipping in patients with exon-3 deletion).

The length of DNA should be described using more conventional and commonly used abbreviation Mb, instead of Mbps.

Author Response

Reviewer 1: 
Comments and suggestions for the authors: 
The research article ijms-899772 by Li et al proposes specific antisense oligomers (AO) as a potential treatment for a sub-type of Parkinson’s disease caused by PRKN exon-3 deletion. It has been shown that removal of exon-3 alters the wild type translational reading frame and is associated with severe clinical symptoms whereas deletion of both exon-3 and 4 preserves the reading frame and causes less severe clinical symptoms. Li et al hypothesize that the severe phenotype due to exon-3 deletions can potentially be lessened by introducing AO-mediated exon-4 skipping in patients carrying the naturally according exon-3 deletion, a clever idea that comes with the inherent limitations and disadvantages of AO-mediated treatment approaches. To evaluate this idea, Li et al performed in vitro experiments on fibroblast cells from a patient affected with autosomal recessive juvenile onset Parkinson due to PRKN exon-3 deletion on one allele and a missense mutation on the other allele. Authors show that, in vitro, they were able to “revert” the wild type reading frame in exon-3 deletion-containing patient fibroblasts by using AO-mediated exon-4 skipping. They also show that the “corrected” transcript produces the expected internally truncated parkin protein capable of performing molecular functions to some extent. Overall, it is a well-designed and thought-out study. The experiments include appropriate positive and negative controls, and results are presented as a well written report. In this reviewer’s opinion this article should be accepted for publication. Here are a few minor suggestions for author’s consideration.
Response: Thank you for your time reviewing our manuscript. Your comments and suggestions are valuable for the manuscript. Detailed responses to each comment are listed below: 

Comment 1: Line 116: This is the first-time manuscript discusses about the missense mutation, at least in the draft provided to this reviewer. Up until this point, only exon-3 deletion is referred. I understand later in the methods section authors have explained that the patient is compound heterozygous for exon-3 deletion and a missense mutation, but it would be helpful for readers to provide this information in the results section before line 116.
Response 1: Thank you for your suggestion. We have now added patient mutation details in Results 2.2 before line 116 “Patient compound heterozygous PRKN mutations: exon 3 deletion on one allele and c.719 C>T missense mutation on the other were confirmed by Sanger sequencing (Supplementary Figure 1)” (page 4, lines 109-111).

Comment 2: Figure 2A: The legend for concentration gradient shown at the bottom right corner of the gel should be moved to someplace else so that it is clearly visible whether or not the Control AO, Untreated, and Negative samples have any band corresponding to exon 3+4 skipping (it is possible, at least theoretically, that the genomic deletion containing exon 3 and surrounding intronic sequences includes distant splice enhancers for exon-4, thus causing some exon-4 skipping in patients with exon-3 deletion).
Response 2: Thank you for this observation. We have now moved the concentration gradient out of the gel as shown in the new Figure 2A.  No additional band corresponding to exon 3+4 skipping was observed for Control AO treated, Untreated and Negative samples.

Comment 3: The length of DNA should be described using more conventional and commonly used abbreviation Mb, instead of Mbps.
Response 3: We have now changed Mbps to Mb.

Reviewer 2 Report

This is an interesting paper demonstrating the feasibility of rescuing a subset of patients with exon 3 deletions in parkin, which truncates the protein, by using oligonucleotides to induce exon 4 skipping, which restores parkin to a slightly shortened form of parkin that retains some functionality. As such, if the authors could demonstrate, using multiple parameters, rescue of mitochondrial function and integrity by this truncated form, then it would pave the way for assessing whether this strategy could have implications for partial rescuing or slowing of the pathology in these patients with a natural exon 3 deletion.

The authors state “Genotype-phenotype studies on Parkin-type ARJP show that patients with genomic deletion of both exons 3 and 4 in PRKN present with milder symptoms than patients with the out-of-frame deletion of exons 3 or 4 [11, 12]. These observations suggest that the parkin isoform missing domains encoded by exons 3 and 4 is semi-functional, and thus provides justification that inducing a PRKN mRNA isoform missing exons 3 and 4 may offer a therapeutic avenue for those ARJP patients with disease-causing mutations involving either exon 3 or 4.”

The concepts here need minor revisions for clarity. I believe the authors are stating that the double deletion of exons 3 and 4 in carriers of an in-frame deletion cause milder symptoms than deletions of either exon 3 or exon 4 alone, when these are out-of-frame deletions. Since out-of-frame deletions of exons 3 or 4 can have major consequences for the protein downstream, then the effects are not limited to exon 3 and 4 alone. Simply adding the word “alone” to the description would already improve the text.

Since there are known pathogenic variants in exons 3 and 4, this would imply that either loss of the normal function of the protein domain encoded by those exons (linker and RING0) is pathogenic, or that a toxic gain-of-function effect occurs. If the former is true, then the logic of deleting exons 3 and 4 providing therapeutic value might not hold, unless the truncated protein retained enough functionality of the RING0 region in combination with the rest of the protein domains to somehow compensate for the deletion of exons 3 and 4. If partially functional parkin maintains enough function from part of RING0 and downstream protein domains to produce milder symptoms in carriers of those particular variations, then it would support further investigation of this avenue. The authors need to demonstrate the partial restoration of functions associated with parkin, which here they have tried to do with recruitment to the depolarised mitochondria, and p53 transcriptional suppression.

The authors used “Parkin-type ARJP patient dermal fibroblasts” in their experiments. It needs to be explicitly stated somewhere (materials and methods etc) the exact genetic background/mutation. From the following phrases “In the untreated samples (Figure. 2A), the amplicons representing the transcript with the exon 3 genomic deletion are in slight excess of the full-length product, presumably due to preferential amplification of the shorter transcript”  and “found to be missing exon 4 from the full-length allele carrying the c.719 C>T missense mutation” – this would imply the authors are using cells from a compound heterozygous mutation carrier, with a truncated exon 3 allele and  a missense mutation (p.Tyr240Met in exon 6) on the other allele. They can just state that clearly in the appropriate section.

The authors state “Low levels of endogenous PRKN exon 4 skipping were also observed in untreated or GTC treated samples. Because skipping of PRKN exon 4 from Δ3 transcripts is expected to restore the PRKN reading frame, we analysed the parkin protein expression in the PMO-treated and untreated samples and found a shortened parkin protein that was most likely translated from ∆3+4 transcripts (Figure. 3B). No truncated protein was observed in the GTC treated or untreated patient-derived fibroblasts.”

However, the bands on the Western blot in 3B are really quite faint, and despite the 20um band showing a clear presence of the truncated protein, a gel with longer exposures (in the supplement maybe) would be necessary to rule out that the low-levels of endogenous exon 4 skipping do not also lead to truncated protein also at low levels.

Figure 3 claiming to show the co-localisation of parkin to the mitochondria is, unfortunately, not convincing due to the very low magnification and lack of resolution of the mitochondrial network within the cells. It should be possible to take images at higher magnification, even focussing on a few cells, to show the diffuse parkin staining migrating to more punctate mitochondrial staining after CCCP treatment. Higher magnifications of mitochondrial networks might also provide an easy source of additional evidence of partial rescue of mitochondrial phenotypes (see comments below). Examples of this are shown in the 2017 paper of Zanon et al (PMID: 28379402).

The central point of the experiments here is to show rescue of parkin function, which should also really focus on mitochondrial integrity and function. The authors mention “maintaining mitochondrial homeostasis” but don’t provide enough evidence of this. While regulation of p53 transcription does address an element of parkin function that could well be related to neurodegeneration, it’s clear that a major role, and the reason parkin is recruited to damaged mitochondria, is the maintenance of mitochondrial intergrity/function. Therefore, in validating this approach, it would be important to not only better demonstrate recruitment to depolarised mitochondria (as per the comments above), but also document some measures of restoration of mitochondrial function. There are several assays to choose from, from assaying mitophagy (via Western blotting, or the mt-Keima assay), or rescuing disrupted complex I activity, or looking at ROS production (with e.g. mitosox), or mitochondrial membrane potential using dyes, or even simply documenting restoration of disrupted mitochondrial networks using higher magnification images (as mentioned previously).  Without better evidence of rescue of mitochondrial function by the restored truncated form of parkin, the case for this strategy as potentially therapeutic is weakened. By contrast, with at least some further evidence of rescue of mitochondrial function, this story will be much stronger. Some of these assays/analyses are relatively easy to perform with the existing resources, and should not take too much time.

While the examples of DMD and Becker MD and the dystrophin gene in the discussion are illustrative, especially for the approach taken – if space is short, the authors could reduce that part of the discussion to focus just on the parkin story.

Given that the approach was largely unsuccessful, I fail to see here the relevance of the attempts to develop a strategy to induce deletion of the exon 3 in normal cells. This distracts somewhat from the main point of this manuscript, which is to develop a potential therapeutic strategy in cases of natural exon 3 deletion, to rescue function by inducing skipping of exon 4 to produce a truncated protein with partial function, enough to rescue or reduce symptoms in patients carrying homozygous or compound heterozygous mutations that include a deleted exon 3 allele. Unless the authors can better justify what such an exon 3 skipping strategy brings to the table in this argument, I would recommend removing it from the manuscript. Currently it reads as though the authors were also doing these experiments alongside the induced exon 4 skipping and restoration of a truncated partially functional form of parkin, and just included them in the manuscript.

Minor language corrections:

“Genomic deletions in PRKN are responsible for 50% of familial early-onset PD cases, with exon 3 and/or 4 deletions (fragile hotspot in 6q26) make up half of these cases.”  - that should be making up, not make up.

“A dose dependent repression of p53 expression” should be dose-dependent repression.

“Slightly higher level of exon 3 skipping (35%) was induced” – should read A slightly higher level, or slightly higher levels.

There may be other examples of such minor linguistic corrections – the authors should proof read carefully a revised manuscript and correct also such minor points.

Author Response

Reviewer 2: 
Comments and suggestions for authors
This is an interesting paper demonstrating the feasibility of rescuing a subset of patients with exon 3 deletions in parkin, which truncates the protein, by using oligonucleotides to induce exon 4 skipping, which restores parkin to a slightly shortened form of parkin that retains some functionality. As such, if the authors could demonstrate, using multiple parameters, rescue of mitochondrial function and integrity by this truncated form, then it would pave the way for assessing whether this strategy could have implications for partial rescuing or slowing of the pathology in these patients with a natural exon 3 deletion.
Response: Thank you for the time and valuable comments and suggestions. We have incorporated the changes into the manuscript according to the comments. Detailed changes to revise this manuscript are listed below.

Comment 1: The authors state “Genotype-phenotype studies on Parkin-type ARJP show that patients with genomic deletion of both exons 3 and 4 in PRKN present with milder symptoms than patients with the out-of-frame deletion of exons 3 or 4 [11, 12]. These observations suggest that the parkin isoform missing domains encoded by exons 3 and 4 is semi-functional, and thus provides justification that inducing a PRKN mRNA isoform missing exons 3 and 4 may offer a therapeutic avenue for those ARJP patients with disease-causing mutations involving either exon 3 or 4.”

The concepts here need minor revisions for clarity. I believe the authors are stating that the double deletion of exons 3 and 4 in carriers of an in-frame deletion cause milder symptoms than deletions of either exon 3 or exon 4 alone, when these are out-of-frame deletions. Since out-of-frame deletions of exons 3 or 4 can have major consequences for the protein downstream, then the effects are not limited to exon 3 and 4 alone. Simply adding the word “alone” to the description would already improve the text.
Response 1: The word “alone” has been added after either exon 3 or exon 4 (page 2, line 55). Thank you for your suggestion. 

Comment 2: Since there are known pathogenic variants in exons 3 and 4, this would imply that either loss of the normal function of the protein domain encoded by those exons (linker and RING0) is pathogenic, or that a toxic gain-of-function effect occurs. If the former is true, then the logic of deleting exons 3 and 4 providing therapeutic value might not hold, unless the truncated protein retained enough functionality of the RING0 region in combination with the rest of the protein domains to somehow compensate for the deletion of exons 3 and 4. If partially functional parkin maintains enough function from part of RING0 and downstream protein domains to produce milder symptoms in carriers of those particular variations, then it would support further investigation of this avenue. The authors need to demonstrate the partial restoration of functions associated with parkin, which here they have tried to do with recruitment to the depolarised mitochondria, and p53 transcriptional suppression.

The authors used “Parkin-type ARJP patient dermal fibroblasts” in their experiments. It needs to be explicitly stated somewhere (materials and methods etc) the exact genetic background/mutation. From the following phrases “In the untreated samples (Figure. 2A), the amplicons representing the transcript with the exon 3 genomic deletion are in slight excess of the full-length product, presumably due to preferential amplification of the shorter transcript”  and “found to be missing exon 4 from the full-length allele carrying the c.719 C>T missense mutation” – this would imply the authors are using cells from a compound heterozygous mutation carrier, with a truncated exon 3 allele and  a missense mutation (p.Tyr240Met in exon 6) on the other allele. They can just state that clearly in the appropriate section.
Response 2: Large deletions make up 50-60% of all Parkin mutations and this number is expected to be even higher (Christine Klein, et.al. Cold Spring Harb Perspect Med. 2012). Among small pathogenic mutations, frame-shifting and nonsense mutations account for a big portion, for example 67.5% (27/40) in the Portuguese population (Sara Morais, et.al. Neurol Genet. 2016.) and 36.4% (4/11) in a Japanese cohort (Nacer Abbas, et.al. Hum Mol Genet. 1999). Some missense mutations in exon 3 and 4, such as p.Ala82Glu in exon 3 and p.Lys161Asn in exon 4 are reported to alter the protein conformation or disrupt ubiquitination properties of parkin (Katja Hedrich, et.al. Hum Mol Genet. 2001; Sathya R Sriram,et.al. Hum Mol Genet. 2005). 
As we discussed in our manuscript, several parkin constructs without the Ubl domain, the first 95 aa, or the Ubl and the following linker domain, were not compromised in their catalytic activity, compared to normal parkin (Jean-Francois Trempe, et.al. Science. 2013) (page 12, lines 262-264). And more importantly, there are genotype-phenotype studies showing patients with both exons 3 and 4 deletion have milder symptoms than patient with the deletion of either exon 3 or 4 alone (William Haylett, et.al. Parkinson’s disease. 2016; Hirohide Asai, et.al. Biochem Biophys Res Commun. 2010). Therefore, we hypothesized that the exon 3 and 4 skipped parkin missing 25% of the Ubl domain, the entire linker domain and 44% of the RING0 domain to be semi-functional and this is validated by our functional assays. 
Initially we put the patient mutation information in the supplementary data, but it is definitely better to state this information in the main text and thank you for your suggestion. We have updated patient mutation details in Results 2.2 (page 2 lines 109-111). 

Comment 3: The authors state “Low levels of endogenous PRKN exon 4 skipping were also observed in untreated or GTC treated samples. Because skipping of PRKN exon 4 from Δ3 transcripts is expected to restore the PRKN reading frame, we analysed the parkin protein expression in the PMO-treated and untreated samples and found a shortened parkin protein that was most likely translated from ∆3+4 transcripts (Figure. 3B). No truncated protein was observed in the GTC treated or untreated patient-derived fibroblasts.”
However, the bands on the Western blot in 3B are really quite faint, and despite the 20um band showing a clear presence of the truncated protein, a gel with longer exposures (in the supplement maybe) would be necessary to rule out that the low-levels of endogenous exon 4 skipping do not also lead to truncated protein also at low levels.
Response 3: A gel image with longer exposures is included in supplementary data (Supplementary Figure 2). There is no band indicating truncated protein from the low-levels of endogenous exon 4 skipping in the GTC treated or untreated patient-derived fibroblasts. 

Comment 4: Figure 3 claiming to show the co-localisation of parkin to the mitochondria is, unfortunately, not convincing due to the very low magnification and lack of resolution of the mitochondrial network within the cells. It should be possible to take images at higher magnification, even focussing on a few cells, to show the diffuse parkin staining migrating to more punctate mitochondrial staining after CCCP treatment. Higher magnifications of mitochondrial networks might also provide an easy source of additional evidence of partial rescue of mitochondrial phenotypes (see comments below). Examples of this are shown in the 2017 paper of Zanon et al (PMID: 28379402).

The central point of the experiments here is to show rescue of parkin function, which should also really focus on mitochondrial integrity and function. The authors mention “maintaining mitochondrial homeostasis” but don’t provide enough evidence of this. While regulation of p53 transcription does address an element of parkin function that could well be related to neurodegeneration, it’s clear that a major role, and the reason parkin is recruited to damaged mitochondria, is the maintenance of mitochondrial intergrity/function. Therefore, in validating this approach, it would be important to not only better demonstrate recruitment to depolarised mitochondria (as per the comments above), but also document some measures of restoration of mitochondrial function. There are several assays to choose from, from assaying mitophagy (via Western blotting, or the mt-Keima assay), or rescuing disrupted complex I activity, or looking at ROS production (with e.g. mitosox), or mitochondrial membrane potential using dyes, or even simply documenting restoration of disrupted mitochondrial networks using higher magnification images (as mentioned previously).  Without better evidence of rescue of mitochondrial function by the restored truncated form of parkin, the case for this strategy as potentially therapeutic is weakened. By contrast, with at least some further evidence of rescue of mitochondrial function, this story will be much stronger. Some of these assays/analyses are relatively easy to perform with the existing resources, and should not take too much time.
Response 4: The cells treated and immunolabelled for Figure 3 were performed in early 2018 and we have lost those slides due to contamination. This manuscript was submitted after a provisional patent was filed in 2020. We have repeated this experiment twice to provide a better image for figure 3, however the CCCP was expired and not able to depolarise the mitochondria. We were unable to get access to the new chemical and due to the current COVID-19 situation, most reagent orders will take 6-8 weeks to arrive. Therefore, we were unable to provide high resolution images for figure 3. However, we are able to provide an image with more cells in the PMO treated group and non-depolarising conditions for both patient and normal cells as suggested by another reviewer to the new Figure 3 using the images captured in 2018. We also provided more figures indicating the colocalization of induced functional parkin protein and depolarised mitochondria as Supplementary Figure 3. 
Previously, we had considered other assays including the mt-Keima assay and looking at the ROS production, however due to funding limitations we have not yet done this study. We will do more assays as suggested in future studies to make to strengthen the data. Thank you for your suggestions. 

Comment 5: While the examples of DMD and Becker MD and the dystrophin gene in the discussion are illustrative, especially for the approach taken – if space is short, the authors could reduce that part of the discussion to focus just on the parkin story.
Response 5: Thank you for your suggestion. This part has been reduced. 

Comment 6: Given that the approach was largely unsuccessful, I fail to see here the relevance of the attempts to develop a strategy to induce deletion of the exon 3 in normal cells. This distracts somewhat from the main point of this manuscript, which is to develop a potential therapeutic strategy in cases of natural exon 3 deletion, to rescue function by inducing skipping of exon 4 to produce a truncated protein with partial function, enough to rescue or reduce symptoms in patients carrying homozygous or compound heterozygous mutations that include a deleted exon 3 allele. Unless the authors can better justify what such an exon 3 skipping strategy brings to the table in this argument, I would recommend removing it from the manuscript. Currently it reads as though the authors were also doing these experiments alongside the induced exon 4 skipping and restoration of a truncated partially functional form of parkin, and just included them in the manuscript.
Response 6: Genomic deletions in PRKN are responsible for 50% of familial early-onset PD cases, with exon 3 and/or 4 deletions (fragile hotspot in 6q26) making up half of these cases. If we can skip exon 4 in patients with exon 3 deletion and exon 3 in patients with exon 4, around 25% of patients can be treated. This was our original focus on developing exon skipping strategies as treatment for Parkinson’s arising from parkin mutations. However, we do not have access to patient-derived fibroblasts carrying exon 4 deletion mutations and therefore we optimised exon 3 skipping using healthy fibroblasts. Although exon 3 skipping using a single AO was not as efficient as skipping exon 4, the skipping efficiency was increased by a cocktail treatment. We could not confirm exon 3 skipping in patient-derived cells, but we think it is valuable to report the similar strategy adopted. We have previously reported “hard-to-skip” exons (type 4 exons) in DMD (Steve Wilton, et.al. Mol Ther. 2007), and we thought it would be interesting to show that this type of exon exists in other genes. Parkin has some superficial similarities to DMD, such as enormous in size, small exon compared to intron component, and prone to deletion mutations. 

Minor language corrections:
Comment 1: “Genomic deletions in PRKN are responsible for 50% of familial early-onset PD cases, with exon 3 and/or 4 deletions (fragile hotspot in 6q26) make up half of these cases.”  - that should be making up, not make up.
Response 1: We have now changed “make up” to “making up”.

Comment 2: “A dose dependent repression of p53 expression” should be dose-dependent repression.
Response 2: We have now changed “dose dependent” to “dose-dependent”.  

Comment 3: “Slightly higher level of exon 3 skipping (35%) was induced” – should read A slightly higher level, or slightly higher levels.
Response 3: “Slightly higher level” has been changed to “A slightly higher level”.

Comment 4: There may be other examples of such minor linguistic corrections – the authors should proof-read carefully a revised manuscript and correct also such minor points.
Response 4: All authors have carefully proof-read the revised manuscript.

Reviewer 3 Report

In general, the organization and of this manuscript is much focused. I thoroughly enjoyed reading the manuscript and have only minor issues be addressed:

  • For temperature, please put a space in between the number and the unit.
  • It should be μL instead of μl, please check throughout the manuscript.

Author Response

Reviewer 3
Comments and suggestions for the authors: 
In general, the organization and of this manuscript is much focused. I thoroughly enjoyed reading the manuscript and have only minor issues be addressed:
For temperature, please put a space in between the number and the unit.
It should be μL instead of μl, please check throughout the manuscript.
Responses: Thank you for reviewing this manuscript. Changes have been made according to your suggestions. 

Reviewer 4 Report

Li et al. tested in vitro the use of antisense oligonucleotides (AOs) for the treatment of ARJP to restore parkin protein function. They were able to induce PRKN exon 4 skipping, in patient-derived fibroblast cells carrying a heterozygous PRKN exon 3 deletion, by using AOs designed against exon splicing enhancers or exon-intron boundaries. The key finding is the identification of an antisense oligonucleotides, AO7, able to generate a 50% exon 4 skipping at the lowest concentration. This was demonstrated at both transcript and protein levels, with the identification by western-blot of a truncated parkin protein of 38kDa with a preserved catalytic RING domain. The authors suggest that the shorter parkin isoform retains functionality, although this last claim is not fully supported by the data. The author also tried to perform a PRKN exon 3 skipping, but this was much less efficient, and required the use of cocktails of multiple antisense oligonucleotides.

The question of the manuscript is original and well defined. Overall, their findings are robust and show the promise of AOs for the treatment of specific ARJP mutations and advance the current knowledge for personalized medicine. The results generated in patient-derived fibroblasts are significant, however further studies are needed to validate the identified AO (AO7) into a closer model of the disease, especially on neuronal cell types known to be involved in the pathology. The authors acknowledge this limitation in the discussion. The study design, data analysis and data presentation are appropriate. The author should temper their conclusion about the functionality of the truncated protein that might require further analysis.

Major comments

  1. For a better understanding of the hypothesis behind the experimental design, the authors should include in the material and methods section a more detailed description of the patient-derived fibroblast cell line, in particular genotype information regarding mutations on the PRKN gene. This info is present throughout the manuscript but would be better if it was summarized and easily accessible for the reader.
  2. The study is well designed, with attention to identifying the most effective AOs and multiple doses of each AOs were tested. For all experiments, the authors should specify how many times the experiment was repeated to ensure reproducibility and include results from appropriate statistical tests directly in the figures, for instance:
    1. In the case of Figure 2D where a densiometric analysis is performed to quantify the exon skipping efficiency, the number of replicates should be sufficient to produce statistical significance.
    2. The fig. 3D should also contain information regarding statistical significance of the downregulation of p53 (usually indicated by asterisks in the figure).

  3. For a safe and effective window of use of the AO, did the authors monitor the cell viability or toxicity over a titration course?
  4. Less convincing is the assessment of the functionality of the resulting truncated protein. In particular, the immunofluorescent assessment of parkin-tomm20 colocalization is problematic. The representative panel for the patient-derived fibroblasts treated with PMO in the presence of CCCP, shows less cells (lower cell density) compared to the other conditions. The colocalization should be quantified by counting the percent of cells showing Parkin colocalization with TOM20 in each condition. In addition, illustrative higher magnification panels should be provided to better show the absence of colocalization at the subcellular scale. Finally, a negative control panel in a nondepolarizing condition would be useful to show the absence of colocalization of the two proteins. This would better support the interesting finding of the 40% downregulation of the parkin target p53 gene obtained by RT-qPCR.

Minor comments

  1. Please check the text at the end of page 5, “Two amplicons of 711bp and 470 bp, corresponding to exon 3 deletion and the skipping of exon 4 from the exon 3-deleted transcript respectively.” The 711bp should correspond to exon 4 skipping.
  2. In Figure 3A, amplicon bands are not clearly labeled. The 711bp exon 4 skipping amplicon label is missing.
  3. The image quality of Figure 1 and Figure 4 should be improved for the publication to be less pixelated and all texts should remain readable.
  4. The western blot analysis (Fig. 3B) provides evidence of the presence of the truncated parkin protein, a result of the PRKN exon 4 skipping, revealing protein molecular weight and relative expression level. The housekeeping gene protein level is not showed.
  5. In figure 3D, the bar plot can be improved by changing the y axis title to “p53 relative expression level (normalized to TBP)” for example.
  6. In Figure 2, the SMN transcript was used as a housekeeping gene. The choice of the housekeeping gene is very important to prove that it is not affected by the treatment. Can the author justify their choice? Why not try using other genes like IPO8 MRPL19, PSMC4 that are most stably expressed in dermal fibroblasts as identified by the most commonly used selection algorithm? Is SMN not a multifunctional protein of motor neuron?

Author Response

Reviewer 4
Comments and suggestions for the authors: 
Li et al. tested in vitro the use of antisense oligonucleotides (AOs) for the treatment of ARJP to restore parkin protein function. They were able to induce PRKN exon 4 skipping, in patient-derived fibroblast cells carrying a heterozygous PRKN exon 3 deletion, by using AOs designed against exon splicing enhancers or exon-intron boundaries. The key finding is the identification of an antisense oligonucleotides, AO7, able to generate a 50% exon 4 skipping at the lowest concentration. This was demonstrated at both transcript and protein levels, with the identification by western-blot of a truncated parkin protein of 38kDa with a preserved catalytic RING domain. The authors suggest that the shorter parkin isoform retains functionality, although this last claim is not fully supported by the data. The author also tried to perform a PRKN exon 3 skipping, but this was much less efficient, and required the use of cocktails of multiple antisense oligonucleotides.

The question of the manuscript is original and well defined. Overall, their findings are robust and show the promise of AOs for the treatment of specific ARJP mutations and advance the current knowledge for personalized medicine. The results generated in patient-derived fibroblasts are significant, however further studies are needed to validate the identified AO (AO7) into a closer model of the disease, especially on neuronal cell types known to be involved in the pathology. The authors acknowledge this limitation in the discussion. The study design, data analysis and data presentation are appropriate. The author should temper their conclusion about the functionality of the truncated protein that might require further analysis.
Response: Thank you for reviewing our manuscript and giving us constructive comments and suggestions. Changes that have been made are listed below under every specific comment. 

Major comments
Comment 1: For a better understanding of the hypothesis behind the experimental design, the authors should include in the material and methods section a more detailed description of the patient-derived fibroblast cell line, in particular genotype information regarding mutations on the PRKN gene. This info is present throughout the manuscript but would be better if it was summarized and easily accessible for the reader.
Response 1: Thank you for your suggestion. We have updated the patient mutation information “Patient compound heterozygous PRKN mutations: exon 3 deletion on one allele and c.719 C>T missense mutation on the other were confirmed by Sanger sequencing (Supplementary Figure 1)” in Results 2.2 (page 4, lines 109-111).

Comment 2: The study is well designed, with attention to identifying the most effective AOs and multiple doses of each AOs were tested. For all experiments, the authors should specify how many times the experiment was repeated to ensure reproducibility and include results from appropriate statistical tests directly in the figures, for instance:
In the case of Figure 2D where a densiometric analysis is performed to quantify the exon skipping efficiency, the number of replicates should be sufficient to produce statistical significance.
The fig. 3D should also contain information regarding statistical significance of the downregulation of p53 (usually indicated by asterisks in the figure).
Response 2: All the experiments were done in three biological replicates. The p value indicating the statistical significance of downregulation of p53 was shown in the figure legend of Figure 3, and we have updated this information in Figure 3D.  

Comment 3: For a safe and effective window of use of the AO, did the authors monitor the cell viability or toxicity over a titration course?
Response 3: In this study two chemistries including 2′-O-methyl AO on a phosphorothioate backbone (2′OMe) and phosphorodiamidate morpholino oligomer (PMO) were used. 2′OMe AOs have known life-threatening toxicities and are ineffective in clinical trials that led to the rejection of AO drug based on this chemistry. However, since it is inexpensive to synthesize, we use 2′OMe AOs for AO sequence screening and the most prominent sequence will be purchased as PMO. PMO is an established clinically safe chemistry that now has three FDA approvals including Eteplirsen, Golodirsen and Vitolarsen. A low level of cell death after the PMO transfection including the PMO targeting PRKN and control PMO from Gene Tools (GTC) using Endoporter We have now added this information in the manuscript (page 5, lines 141-143). This is a pilot study to validate the idea of the exon skipping strategy, so currently it is not our purpose to determine the effective window of our PMO to restore parkin function in vitro; however in further in vivo studies this will definitely be done before going to human trials.    

Comment 4: Less convincing is the assessment of the functionality of the resulting truncated protein. In particular, the immunofluorescent assessment of parkin-tomm20 colocalization is problematic. The representative panel for the patient-derived fibroblasts treated with PMO in the presence of CCCP, shows less cells (lower cell density) compared to the other conditions. The colocalization should be quantified by counting the percent of cells showing Parkin colocalization with TOM20 in each condition. In addition, illustrative higher magnification panels should be provided to better show the absence of colocalization at the subcellular scale. Finally, a negative control panel in a nondepolarizing condition would be useful to show the absence of colocalization of the two proteins. This would better support the interesting finding of the 40% downregulation of the parkin target p53 gene obtained by RT-qPCR.
Response 4: The cells treated and immunolabelled for Figure 3 was performed in 2018 and we have lost those slides due to contamination. This manuscript was submitted after a provisional patent is filed. We have repeated this experiment several times, however the CCCP was expired and not able to depolarise the mitochondria. Due to the current COVID-19 situation, new reagents will take 6-8 weeks to arrive. Therefore, we were unable to provide high resolution images to the figure. However, we provide an image with more cells in the PMO treated group and non-depolarising conditions for both patient and normal cells as suggested by another reviewer to the new Figure 3 using the images captured in 2018. We also provided more figures indicating the colocalization of induced functional parkin protein and depolarised mitochondria as Supplementary Figure 3. 

Minor comments
Comment 1: Please check the text at the end of page 5, “Two amplicons of 711bp and 470 bp, corresponding to exon 3 deletion and the skipping of exon 4 from the exon 3-deleted transcript respectively.” The 711bp should correspond to exon 4 skipping.
Response 1: Thank you for pointing this out. The 711bp band corresponds to the skipping of exon 4 from the full-length transcript. We have now made this change in the manuscript. 

Comment 2: In Figure 3A, amplicon bands are not clearly labelled. The 711bp exon 4 skipping amplicon label is missing.
Response 2: We have now added the label for the 711bp exon 4 skipping amplicon. 

Comment 3: The image quality of Figure 1 and Figure 4 should be improved for the publication to be less pixelated and all texts should remain readable.
Response 3: The splicing motif images in Figure 1 and Figure 4 are downloaded from the SpliceAid website and we have tried to get the resolution as high as we can. We have added notes and links for these images in the figure legends for the readers to check the original images. 

Comment 4: The western blot analysis (Fig. 3B) provides evidence of the presence of the truncated parkin protein, a result of the PRKN exon 4 skipping, revealing protein molecular weight and relative expression level. The housekeeping gene protein level is not showed.
Response 4: We have now included the gel image of housekeeping protein β-tubulin.

Comment 5: In figure 3D, the bar plot can be improved by changing the y axis title to “p53 relative expression level (normalized to TBP)” for example.
Response 5: Thank you for your suggestions. We have now changed the title of y axis to p53 relative expression level. 

Comment 6: In Figure 2, the SMN transcript was used as a housekeeping gene. The choice of the housekeeping gene is very important to prove that it is not affected by the treatment. Can the author justify their choice? Why not try using other genes like IPO8 MRPL19, PSMC4 that are most stably expressed in dermal fibroblasts as identified by the most commonly used selection algorithm? Is SMN not a multifunctional protein of motor neuron?
Response 6: SMN is a functional protein of motor neuron, however SMN is consistently expressed in many cell lines including human fibroblast. SMN alternative transcripts are affected by cell stress and splicing events. By using SMN as housekeeping gene, we can generally determine if the cells are under stress after the antisense oligonucleotide (AO) treatment. Therefore, we routinely use SMN as a housekeeping gene.